# Abnormal accumulation of lipid droplets in neurons induces the conversion of alpha-Synuclein to proteolytic resistant forms in a *Drosophila* model of Parkinson's disease

Victor Girard[1], Florence Jollivet[1], Oskar Knittelfelder[2], Marion Celle[1], Jean-Noel Arsac[3], Gilles Chatelain[1], Daan M. Van den Brink[1,4], Thierry Baron[3], Andrej Shevchenko[2], Ronald P. Kühnlein[5,6,7], Nathalie Davoust[1*‡], Bertrand Mollereau[1*‡]

1 Laboratory of Biology and Modelling of the Cell, UMR5239 CNRS/ENS de Lyon, INSERM U1210, UMS 3444 Biosciences Lyon Gerland, University of Lyon, Lyon, France, 2 Max-Planck-Institute of Molecular Cell Biology and Genetics, Dresden, Germany, 3 Neurodegenerative Disease Unit; French Agency for Food, Environmental and Occupational Health & Safety Laboratory (Anses) of Lyon, University of Lyon, Lyon, France, 4 Plant Systems Physiology, Radboud Institute for Biological and Environmental Sciences, Radboud University, Nijmegen, The Netherlands, 5 Institute of Molecular Biosciences, University of Graz, Graz, Austria, 6 BioTechMed-Graz, Graz, Austria, 7 Field of Excellence BioHealth—University of Graz, Graz, Austria

‡ These authors are co-senior authors on this work.
* nathalie.davoust-nataf@ens-lyon.fr (ND); bertrand.mollereau@ens-lyon.fr (BM)

## Abstract

Parkinson's disease (PD) is a neurodegenerative disorder characterized by alpha-synuclein (αSyn) aggregation and associated with abnormalities in lipid metabolism. The accumulation of lipids in cytoplasmic organelles called lipid droplets (LDs) was observed in cellular models of PD. To investigate the pathophysiological consequences of interactions between αSyn and proteins that regulate the homeostasis of LDs, we used a transgenic *Drosophila* model of PD, in which human αSyn is specifically expressed in photoreceptor neurons. We first found that overexpression of the LD-coating proteins Perilipin 1 or 2 (dPlin1/2), which limit the access of lipases to LDs, markedly increased triacylglycerol (TG) loaded LDs in neurons. However, dPlin-induced-LDs in neurons are independent of lipid anabolic (diacyl-glycerol acyltransferase 1/midway, fatty acid transport protein/dFatp) and catabolic (brummer TG lipase) enzymes, indicating that alternative mechanisms regulate neuronal LD homeostasis. Interestingly, the accumulation of LDs induced by various LD proteins (dPlin1, dPlin2, CG7900 or Klarsicht[LD-BD]) was synergistically amplified by the co-expression of αSyn, which localized to LDs in both *Drosophila* photoreceptor neurons and in human neuroblastoma cells. Finally, the accumulation of LDs increased the resistance of αSyn to proteolytic digestion, a characteristic of αSyn aggregation in human neurons. We propose that αSyn cooperates with LD proteins to inhibit lipolysis and that binding of αSyn to LDs contributes to the pathogenic misfolding and aggregation of αSyn in neurons.

**Data Availability Statement:** All relevant data are within the manuscript and its Supporting Information files.

**Funding:** This work was supported by a grant from the Association France Parkinson to BM. VG salary was supported by the Fondation pour la Recherche Médicale (grant FDT201904008373), ENS Fond Recherche, Servier Research Institute, and the Laboratory of Modelling and Biology of the Cell. AS was supported by Deutsche Forschungsgemeinschaft (grant FOR 2682). RKP was supported by Universität Graz. The funders had no role in study design, data collection and analysis, decision to publish, or preparation of the manuscript.

**Competing interests:** The authors have declared that no competing interests exist.

## Author summary

Parkinson's disease (PD) is a neurodegenerative disease characterized by the neurotoxic aggregation of the alpha-synuclein (αSyn) protein. Cellular models of the disease are also associated with an abnormal fat storage in the form of lipid droplets (LDs). However, in which cells, neuron or glial cells, LDs accumulate in the organism remains unknown. To understand the relationship between αSyn and the accumulation of LDs, we used a *Drosophila* (fruit fly) model of PD. We found that, in the presence of a protein that coats LDs, perilipin, LDs accumulate in photoreceptor neurons of the fly. Interestingly, the accumulation of LDs induced by perilipin or other LD-coating proteins was enhanced in the presence of αSyn. Using human neuronal cell lines and the fly, we could show that LD-coating and αSyn proteins localize at the surface of LDs. Finally, we observed that the process of αSyn aggregation was enhanced in the presence of LDs by using a biochemical approach. We thus propose that the association of αSyn with LDs could contribute to αSyn aggregation and progression of the pathology.

## Introduction

Lipids play crucial roles in many essential cellular functions, including membrane formation, energy production, intracellular and intercellular signal transduction, and regulation of cell death. Fatty acids (FAs) taken up into or synthesized within cells are stored in discrete organelles known as lipid droplets (LDs), which consist of a core of neutral lipids predominantly triacylglycerols (TGs) and sterol esters, surrounded by a monolayer of phospholipids containing numerous LD proteins [1]. Maintenance of LD homeostasis in adipose tissue and in the central nervous system, among other tissues, has emerged as a central process for organismal health, and its dysregulation contributes to many human diseases, such as obesity, atherosclerosis, fatty liver disease, and neurodegenerative disorders such as Parkinson's disease (PD) [2–5].

The mechanisms by which fat is stored and remobilized in LDs are dependent on evolutionarily conserved canonical anabolic and catabolic enzymes [6]. LD biogenesis is initiated at the endoplasmic reticulum membrane, where lipogenesis enzymes, such as *Drosophila* diacylglycerol acyltransferase 1 (DGAT1), encoded by the *midway (mdy)* gene, catalyze the rate limiting step of TG synthesis [7,8]. Also, Fatty acid transport protein 1 (dFatp in *Drosophila*), which functions in a complex with diacylglycerol acyltransferase 2 (DGAT2), promotes LD expansion in *C. elegans*, *Drosophila* and mammalian cells [9–11]. Several other components have also been identified, such as Seipin proteins which form a ring-link structure that facilitates the flow of TGs within LDs [12,13]. In contrast, lipolysis is catalysed by lipases, such as the central and evolutionarily conserved TG lipase Brummer in the fly (Bmm also called dATGL), ortholog of mammalian adipose triglyceride lipase (ATGL) [14].

Lipase activity is controlled by a family of LD proteins called perilipins (PLINs) that play different roles in LD homeostasis, including maintaining LD integrity, limiting basal lipolysis, and interacting with mitochondria [15]. The human genome encodes five PLIN proteins, PLIN1–5 [15–17], whereas the *Drosophila* genome encodes two PLINs, Lsd-1 and Lsd-2 (hereafter named dPlin1 and dPlin2) that play distinct roles in LD homeostasis [6]. dPlin2 is indeed considered as the guardian of LDs by shielding LD away from lipases, while dPlin1 by interacting directly with lipases can either stimulate or inhibit lipolysis [7,18–20]. *dPlin1* expression, like that of human *PLIN1*, is mainly restricted to adipose tissue, but it was also found expressed

in *Drosophila* wing imaginal discs [21]. In contrast *dPlin2* and human *PLIN2* and *3* are expressed ubiquitously [16].

While the lipid storage function of LDs is well understood, less is known about the non-lipid storage functions of LDs, such as their involvement in the regulation of cellular stress and protein handling, folding, and turnover [1,22]. This situation has improved in the last few years; for example, studies with *Drosophila* and vertebrate cellular models have begun to unravel the pathophysiological roles of LDs in regulating stress in cells of the nervous system. Oxidative stress exposure or excitotoxicity induce glial LD accumulation in developing or adult *Drosophila* brain and in mouse glial cells co-cultured with neurons [10,23–26]. In *Drosophila* larvae subjected to hypoxia, LD accumulation in glial cells is thought to play a protective role by enabling relocation of lipids sensitive to peroxidation, such as polyunsaturated FAs, from membrane phospholipids to TGs in the LD core [23]. In contrast to glial cells, LDs are rarely detected in neurons and little is known about their potential pathophysiological relevance [10,27]. Furthermore, LD biogenesis in glia, like in adipose tissue cells, depends on canonical enzymes such as DGAT and Fatp [10,23,25], while the mechanisms regulating their turnover in neurons are unknown.

PD is characterized by the neuronal accumulation of misfolded proteins, including α-synuclein (αSyn), in cytoplasmic aggregates known as Lewy bodies [28,29]. αSyn is a vertebrate-specific 14-kDa presynaptic protein and contains an N-terminal domain consisting of repeated sequences of 11 amino acids that fold into an amphipathic helix upon lipid binding [30,31]. Although the physiological function of αSyn is still unclear, several lines of evidence indicate that αSyn binding to phospholipid membranes is important for vesicle dynamics at the synapse [32]. And a recent study showed that the docking of synaptic vesicles to presynaptic membranes by αSyn depends on lipid composition [33]. Regarding PD, multiple lines of evidence indicate that lipid dysregulations are associated with the disease. In particular, several genome wide association studies have identified genes regulating lipid metabolism as PD risk factors [34]. Also, mutations of glucocerebrosidase (GCase) gene, associated with a decrease in GCase activity, is leading to the accumulation of both glucosylceramide and αSyn and is the highest risk factor of developing PD [35]. In a *Drosophila* model carrying a GCase mutation and expressing human αSyn, it was proposed that the binding of lipids to αSyn contributes to its pathogenic conversion [36]. Furthermore, in yeast and mammalian cellular models of PD, the overexpression of αSyn leads to neutral lipids accumulation and LD formation [37–39]. It was initially proposed that αSyn could promote LD formation by inhibiting lipolysis at the surface of LDs [37]. This hypothesis is supported by several studies showing that αSyn binds to LDs in mammalian cell cultures and to synthetic LDs leading to the assumption that αSyn, similarly to perilipins, is as a *bonafide* LD protein [37,40,41]. Alternatively, it was proposed that αSyn expression induces the accumulation of oleic acid generated by a stearoyl-CoA-desaturase enzyme, which fuels the synthesis of diacylglycerols (DGs) and TGs to promote LD formation [38]. However, there is currently no animal model of PD, in which LD accumulation was observed preventing the investigation of a putative bidirectional interplay between αSyn and LDs *in vivo*.

In the present study, we used a *Drosophila* model in which the neuronal expression of human αSyn has proven to be useful to study the pathological mechanisms of PD [42–44]. We investigated the effects of PLIN and αSyn expression on LD formation in photoreceptor neurons; the role of canonical mechanisms regulating LD metabolism; the subcellular co-localization of αSyn with LDs in *Drosophila* photoreceptors neurons and in human neuroblastoma cells; the potential contribution of αSyn with several LD proteins to the inhibition of lipolysis and the subsequent LD formation; and the potential effects of αSyn–LD binding on the susceptibility of αSyn to misfolding in the context of PD.

## Results

### Distinct anabolic and catabolic mechanisms regulate LD turnover in neurons and in glial cells

LDs are commonly observed in glial cells but rarely in neurons under physiological or pathological conditions [10,23,25]. We hypothesized that the apparent lack of LD might be due to active lipolysis in neurons. We thus tested if the neuronal overexpression of PLIN proteins, which are regulating the access of lipases to LDs [17], could promote the neuronal accumulation of LDs. We focused on photoreceptor neurons of the *Drosophila* eye as a model. The compound eye of *Drosophila* is composed of 800 ommatidia, each containing six outer and two inner photoreceptor neurons surrounded by retina pigment/glial cells (S1A, S1B and S1C Fig) [45]. *dPlin1*::*GFP* [7] and *dPlin2*::*GFP* [20] were expressed in outer photoreceptor neurons of

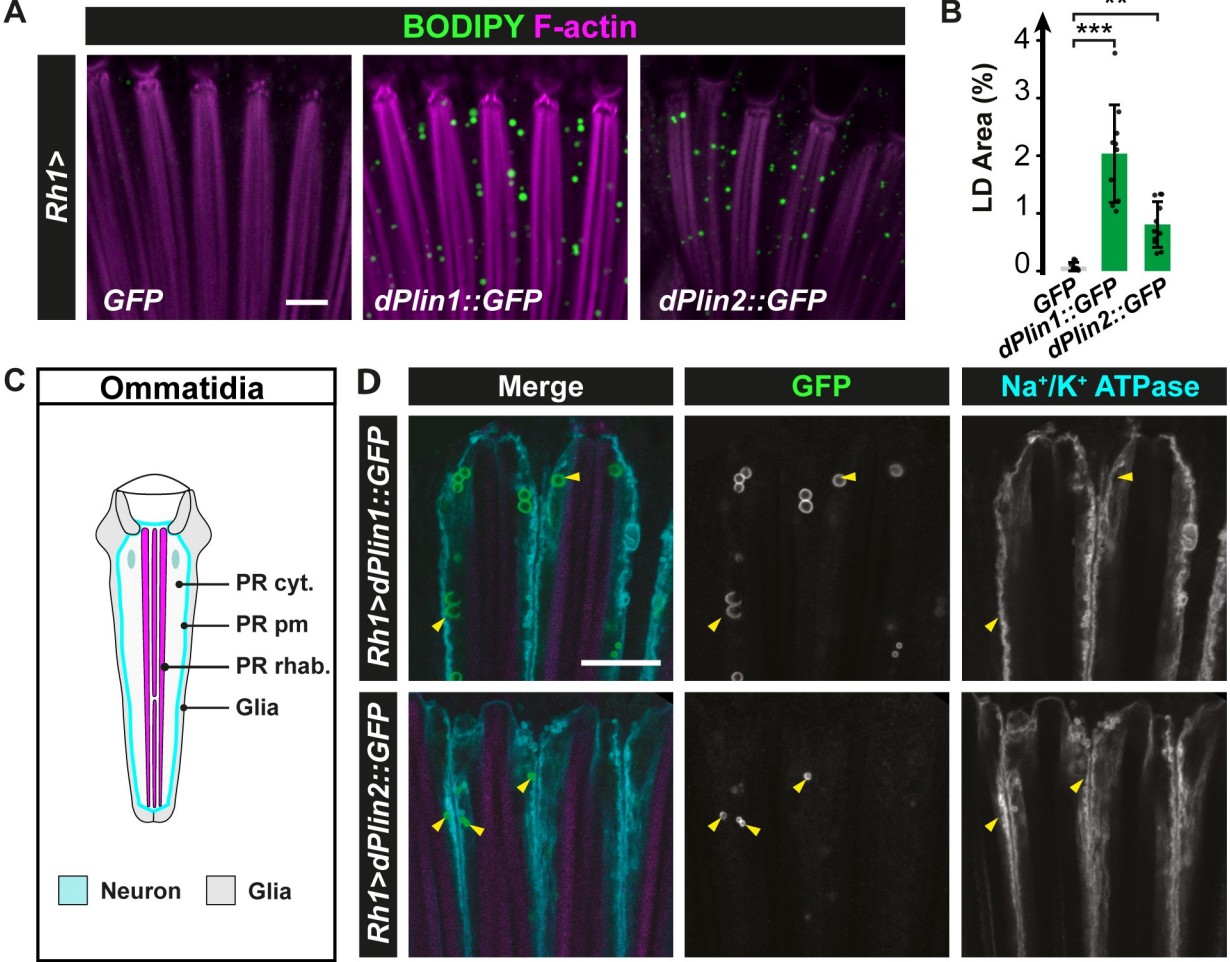

**Fig 1. dPlin1 and dPlin2 promote LD accumulation in *Drosophila* photoreceptor neurons. (A)** LD staining of whole-mount retinas from flies expressing *GFP* (control), dPlin1::GFP, or dPlin2::GFP in photoreceptor neurons (*Rh1-GAL4*). LDs are in green (BODIPY) and photoreceptor rhabdomeres are in magenta (phalloidin-rhodamine). Scale bar, 10 μm. **(B)** Quantification of LD area expressed as % of total retinal area. Data are from the images shown in (A). Mean ± SD. *** p<0.001 by ANOVA with Tukey's HSD test. **(C)** Diagram of a longitudinal section of one ommatidium. Each ommatidium is composed of 8 photoreceptor neurons delimited by a Na$^+$/K$^+$ ATPase positive plasma membrane (cyan), each containing one rhabdomere (magenta), and surrounded by 9 glial cells (also known as retinal pigment cells; light gray). **(D)** Immunostaining of whole-mount retinas from flies expressing dPlin1::GFP or dPlin2::GFP in photoreceptor neurons (*Rh1-GAL4*). Photoreceptor plasma membranes are in cyan (anti-Na$^+$/K$^+$ ATPase) and rhabdomeres are in magenta (phalloidin-rhodamine). dPlin1 and dPlin2 are visible as ring shapes in the photoreceptor cytoplasm (yellow arrowheads). Scale bar, 10 μm.

the adult *Drosophila* using a *rhodopsin 1* (*Rh1*) driver ([46] and S1D Fig). The abundance of LDs was measured by labeling whole-mount retinas with the lipophilic fluorescent dye BOD-IPY. This analysis revealed that, while LDs were virtually undetectable in the retina of 20-day-old control flies, dPlin1::GFP or dPlin2::GFP overexpression led to accumulation of LDs (measured as the percentage of the retina area stained with BODIPY, Fig 1A and 1B). Next, we determined whether this effect of dPlin expression was specific to neurons or also observed in surrounding glial cells due to a non-autonomous effect. We thus immunostained the retina for the Na$^+$/K$^+$ ATPase α subunit, a marker of the photoreceptor plasma membrane [47], which localization is depicted in cyan in the ommatidia diagram (Fig 1C). As shown in Fig 1D, dPlin1::GFP and dPlin2::GFP labelings were visible as rings, characteristic of proteins associated with the LD surface, located within the cytoplasm of photoreceptors but not in the adjacent glial cells. These results indicate that *Rh1*-driven dPlin overexpression, leads to accumulation of LDs in photoreceptor neurons only.

We then asked if LD biogenesis in neurons requires Mdy and dFatp, two canonical enzymes of the TG synthesis [10,23,25]. Photoreceptor neuron-specific knockdown of either *dFatp* or *mdy* had no effect on LD accumulation in dPlin1::GFP-expressing flies (Fig 2A and 2B). To confirm these results, we performed loss of *dFatp* and *mdy* function analyses in *dPlin*-expressing flies. *dPlin1*- or *dPlin2*-induced LDs in photoreceptors were not impacted neither by the loss of *dFatp* (*dFatp*[k10307]) in FLP/FRT-mediated mutant clones nor in global *mdy* (*mdy*[QX25]) mutants (Figs 2C, 2D, 2E, S2A and S2B). These results indicate that dPlin-induced LD accumulation in photoreceptor neurons occurs through a mechanism independent of *dFatp*- and *mdy*-mediated *de novo* TG synthesis and is thus distinct from the mechanism of LD accumulation in glial cells [10,23,25].

Because Perilipins are known to protect LDs from lipase-mediated lipolysis [48], we examined whether the loss of lipolysis could be responsible for the accumulation of LDs. For that we first depleted the main TG lipase Bmm, by RNAi interference, using the glial-specific (*54C-GAL4*) or the photoreceptor-specific *(Rh1-GAL4)* drivers. Knockdown of *bmm* in glial cells resulted in LD accumulation (Fig 2F). In contrast, knockdown of *bmm* did not increase LD abundance in wild type or *dPlin2*-expressing photoreceptors, suggesting that Bmm-mediated lipolysis does not influence LD homeostasis in photoreceptor neurons (Figs 2F, 2G, 2H and S3A). These results were confirmed by observing that LD accumulate in glial but not in photoreceptor cells in *bmm* mutant retina or in a pan-retinal *bmm* knockdown (Figs 2I, S3A and S3B). The fact that LD degradation in *Drosophila* photoreceptor neurons does not involve Bmm, the main *Drosophila* TG lipase, is in contrast with what is observed in other tissues [14]. It suggests that an unknown lipase regulates the degradation of LDs in photoreceptors.

## αSyn synergizes with Perilipin protein to induce LD accumulation in neurons

Having established that Plin levels regulate LD accumulation in neurons, we asked whether the expression of αSyn, which was proposed to interact with LDs as a *bonafide* LD protein [37,40,41], affected LD accumulation in photoreceptor neurons as well. For this, we employed transgenic *Drosophila* lines expressing wild-type human αSyn [42–44,49–51] alone or in combination with dPlin1::GFP or dPlin2::GFP. Notably, while photoreceptor neuron-specific expression of *αSyn* did not induce significant LD accumulation compared with control (LacZ) flies, concomitant expression of *αSyn* and dPlin2::GFP resulted in a striking synergistic effect. Indeed, it resulted in the tripling of the abundance of LDs in photoreceptors compared with either *αSyn* or *dPlin2*::*GFP* expression alone (Fig 3A, 3B and 3C). This result was confirmed using independent fly lines carrying *UAS-dPlin2*::*GFP* and *UAS-αSyn* transgenes inserted at a

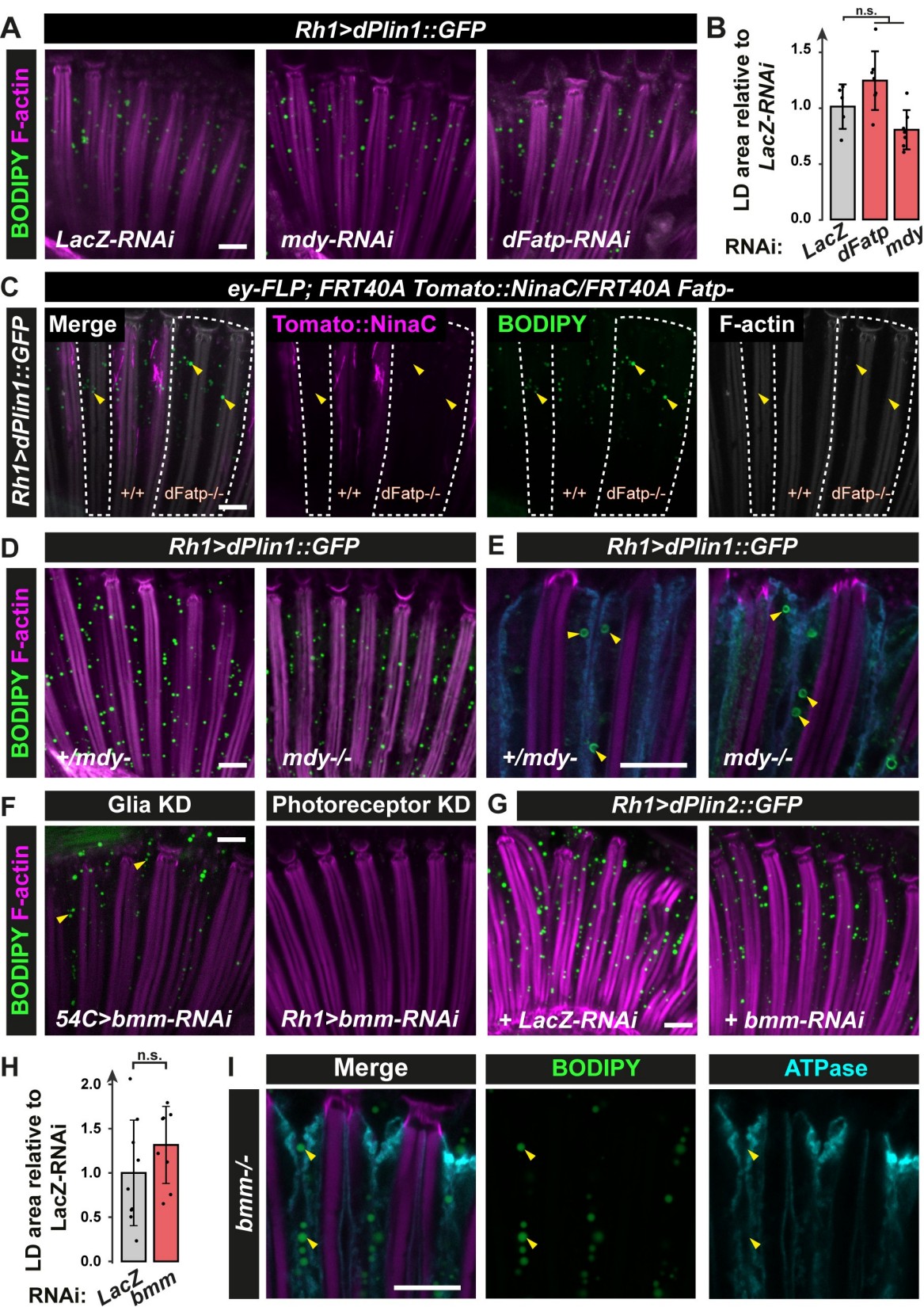

**Fig 2. dPlin1-induced LDs in photoreceptor neurons do not require canonical enzymes involved in TG synthesis (dFatp, Mdy) or degradation (Bmm).** **(A)** LD staining of whole-mount retina from flies with photoreceptor neuron-specific (*Rh1-GAL4*) expression of dPlin1::GFP and *LacZ-RNAi* (control), *dFatp-RNAi*, or *mdy-RNAi*. LDs are in green (BODIPY) and photoreceptor rhabdomeres are in magenta (phalloidin-rhodamine). Scale bar, 10 μm. **(B)** Quantification of LD area from the images shown in (A). Mean ± SD. n.s., not significant by ANOVA. **(C)** LD staining of whole-mount retinas from flipase mediated FRT *dFatp[k10307]* mutant clone in conjunction with expression of dPlin1::GFP in photoreceptor (*Rh1-GAL4*). LDs are in green (BODIPY), wild type photoreceptors are in magenta (*FRT40A-Tomato[ninaC]*) and rhabdomeres are in grey (phalloidin-rhodamine). LDs are visible in photoreceptors lacking *dFatp* (mutant clone surrounded by dashed line). Scale bar, 10 μm. **(D, E)** whole-mount retinas from *mdy[QX25]* heterozygous and homozygous flies expressing dPlin1::GFP in photoreceptor (*Rh1-GAL4*). **(D)** LDs are in green (BODIPY), and rhabdomeres are in magenta (phalloidin-rhodamine). **(E)** Photoreceptor plasma membranes are in cyan (anti-Na⁺/K⁺ ATPase) and rhabdomeres are in magenta (phalloidin-rhodamine). **(F)** LD staining of whole-mount retinas from flies expressing *bmm-RNAi* in photoreceptor (*Rh1-GAL4*) or retina glia (*54C-GAL4*). LDs are in green (BODIPY), and rhabdomeres are in magenta (phalloidin-rhodamine). Scale bar, 10 μm. **(G)** LD staining of whole-mount retinas from flies expressing *dPlin2::GFP* in conjunction with RNAi targeting *LacZ* or *bmm* in photoreceptor (*Rh1-GAL4*). LDs are in green (BODIPY), and rhabdomeres are in magenta (phalloidin-rhodamine). Scale bar, 10 μm. **(H)** Quantification of LD area from the images shown in (G). Mean ± SD. n.s., not significant by t-test. **(I)** Immunostaining of whole-mount retinas from homozygous *bmm* mutant flies. Photoreceptor plasma membranes are in cyan (anti-Na⁺/K⁺ ATPase) and rhabdomeres are in magenta (phalloidin-rhodamine). Scale bar, 10 μm.

different chromosomal sites on second and third chromosomes (S4A and S4B Fig). Finally, we could also observed a synergistic accumulation of LDs induced with *αSyn* and dPlin1::GFP (S5 Fig). Thus, while *Drosophila* photoreceptor neurons contain few LDs under normal physiological conditions, the co-expression of *dPlin1* or *dPlin2* with *αSyn* increased LD content in a synergistic manner. These results suggest that *αSyn* functions in a Plin-like manner and could shield LDs, thus limiting the access of lipases and LD lipolysis.

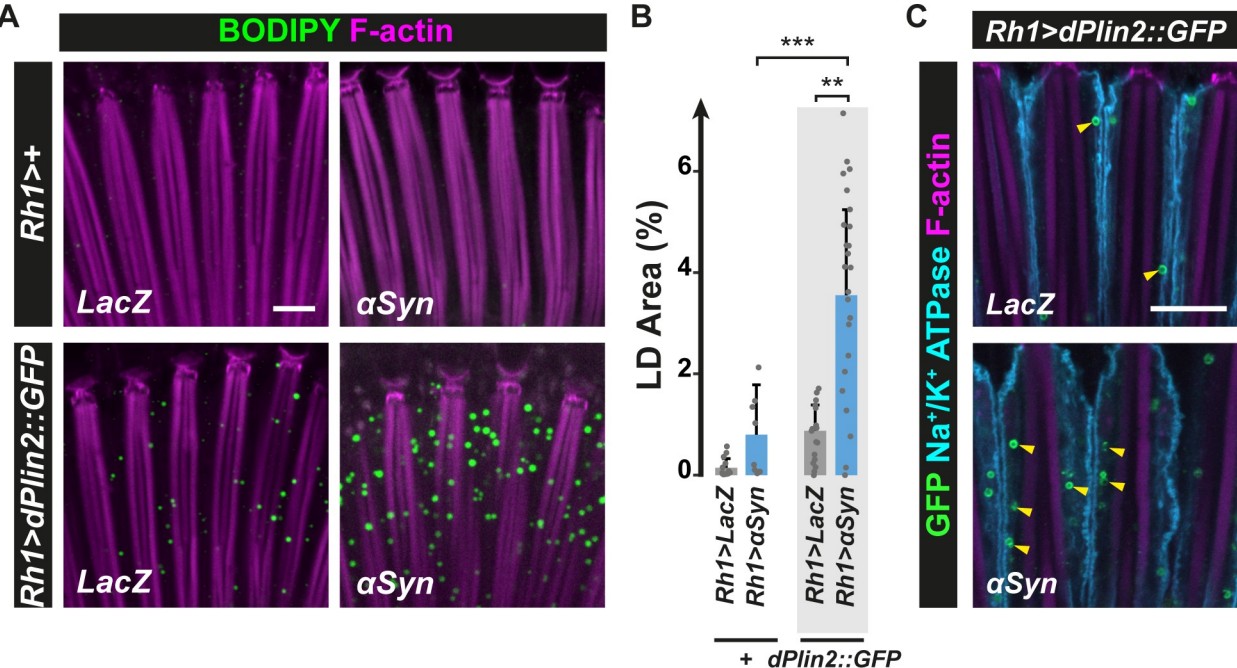

**Fig 3. αSyn enhances dPlin2-induced LD accumulation in *Drosophila* photoreceptor neurons.** **(A)** LD staining of whole-mount retinas from flies expressing LacZ (control) or human αSyn alone or in conjunction with dPlin2::GFP in photoreceptor neurons (*Rh1-GAL4*). LDs are in green (BODIPY) and photoreceptor rhabdomeres are in magenta (phalloidin-rhodamine). Scale bar, 10 μm. **(B)** Quantification of LD area expressed as % of total retinal area. Data are from the images shown in (A). Mean ± SD. ** p<0.01, ***p<0.001 by ANOVA with Tukey's HSD test. **(C)** Immunostaining of whole-mount retinas from flies expressing dPlin2::GFP in photoreceptor neurons (*Rh1-GAL4*). Photoreceptor plasma membranes are in cyan (anti-Na⁺/K⁺ ATPase immunostaining) and rhabdomeres are in magenta (phalloidin-rhodamine labeling of F-actin). dPlin2 staining is visible as ring shapes in the photoreceptor cytoplasm (yellow arrowheads). Scale bar, 10 μm.

## αSyn and Perilipins co-localize at the LD surface in *Drosophila* photoreceptor neurons and in human neuroblastoma cells

αSyn was found to bind LDs in several cellular models of PD in which *αSyn* was overexpressed [37,38], but there is still no evidence that *αSyn* binds LDs in neurons in a model organism of PD. In addition, a putative binding of endogenous αSyn to LDs remains to be shown. We first performed immunostaining of αSyn in flies with photoreceptor neuron-specific expression of dPlin2::GFP and αSyn, which revealed co-localization of the αSyn and dPlin2::GFP at the LD surface (Fig 4A). In addition, we examined protein co-localization in the human neuroblastoma cell line SH-SY5Y transfected or not with *αSyn*. Transfected αSyn co-localized with endogenous PLIN3 (also known as TIP47), which is broadly expressed in human brain cells [52], around circular vesicles, as detected using high-resolution Airyscan microscopy (Fig 4B). Interestingly, αSyn distribution was not uniform and localized in subdomains at the LD surface. In non-transfected SH-SY5Y cells, the endogenous αSyn is expressed at low levels [53], we thus enhanced endogenous αSyn detection by performing proximity ligation assays (PLAs), in which oligonucleotide-coupled secondary antibodies generate a fluorescent signal when the two target protein-bound primary antibodies are in close proximity [54]. Confocal microscopy of non-transfected SH-SY5Y cells labeled with primary antibodies against αSyn and PLIN3 revealed PLA signals that localized around BODIPY-positive structures, indicating that endogenous αSyn and PLIN3 proteins co-localize at the surface of LDs (Figs 4C, S6A and S6B). A more robust PLA signal was observed in SH-SY5Y cells transfected with αSyn confirming the cellular localization of αSyn at the LD surface (Figs 4D, S6C and S6D). Taken together, these experiments show that αSyn co-localizes at the LD surface with the LD-binding protein dPlin2::GFP in *Drosophila* photoreceptor neurons and with PLIN3 in human neuroblastoma cells.

## The accumulation of LDs induced by αSyn and CG7900 is not regulated by lipophagy

Plin and αSyn overexpression promotes LD accumulation in a synergistic manner in *Drosophila* photoreceptor neurons. The fact that αSyn and Plins co-localize at the surface of LDs raises the possibility that Plin-specific coating of LDs precedes αSyn recruitment to the LD surface. Alternatively, Plin-independent LD formation could provide the physical platform, for αSyn to bind and to enhance LD accumulation. To test the latter, we asked if other LD proteins could promote LD accumulation synergistically with αSyn as well. We first characterized the cellular localization of CG7900, the *Drosophila* ortholog of FA amide hydrolase FAAH2, which contains a functional LD-binding domain [55] and that we named *Drosophila* Faah2 (dFaah2). For that we used the larval fat body, which contains diffentatially sized LDs covered by a complex proteome [18] including dPlins [7]. We thus generated a *Drosophila* transgenic UAS-CG7900::GFP line that we first expressed under the control of the larval fat body driver (*r4-GAL4*). We observed a clear accumulation of CG7900::GFP at the LD surface (Fig 5A), which co-localized with dPlin1 as a ring around LDs (Fig 5B). To further assess CG7900 localization at the LD in a cellular context, where LDs are less densely packed, we heterologously expressed a CG7900::RFP in HeLa cells in which endogenous LDs were visualized with BODIPY staining (Fig 5C). We observed a clear co-localization of CG7900 with LDs. Collectively these results substantiate our observation that CG7900 is a *bona fide* LD binding protein. We next asked if *CG7900* overexpression is sufficient to induce the accumulation of LDs in photoreceptors. We observed that indeed the overexpression of CG7900::GFP using the *Rh1* photoreceptor driver led to ring shaped vesicles in the photoreceptor cytoplasm, very similar to the expression of dPlin, suggesting an accumulation of LDs (Fig 5D). Interestingly the fact that the

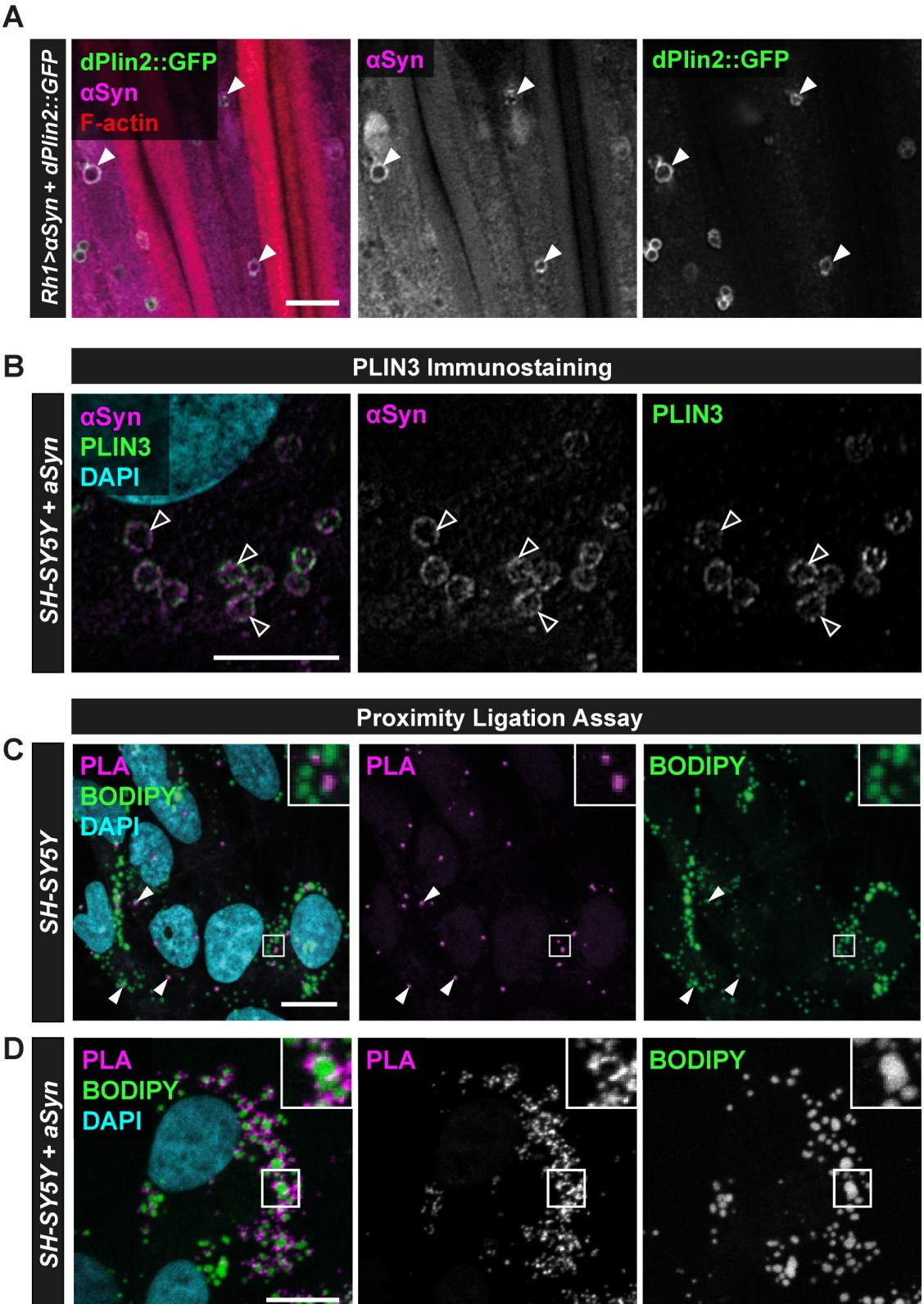

**Fig 4. αSyn co-localizes with PLINs at the surface of LDs in *Drosophila* photoreceptor neurons and in human neuroblastoma cells. (A)** Immunostaining of whole-mount retinas from flies expressing αSyn and dPlin2::GFP in photoreceptor neurons (*Rh1-GAL4*). αSyn is in magenta (anti-αSyn) and photoreceptor rhabdomeres are in red (phalloidin-rhodamine). White arrowheads

indicate co-localization of αSyn and dPlin2 at LDs. Scale bar, 5μm. **(B)** High-resolution Airyscan micrograph of SH-SY5Y neuroblastoma cells transfected with αSyn. αSyn and PLIN3 immunostainings are shown in magenta and green, respectively. Nuclei are counterstained with DAPI (cyan). Arrowheads indicate co-localization of αSyn and PLIN3 staining on LDs. Scale bar, 5 μm. **(C)** Proximity ligation assay between PLIN3 and endogenous αSyn in SH-SY5Y. The PLA signal generated by close proximity of the two protein-bound primary antibodies is shown in magenta, LDs are in green (BODIPY), and nuclei are counterstained with DAPI (cyan). Scale bars, 10 μm. **(D)** Proximity ligation assay between αSyn and PLIN3 in SH-SY5Y cells transfected with human αSyn. The PLA signal is shown in magenta, LDs are in green (BODIPY), and nuclei are counterstained with DAPI (cyan). Scale bars, 10 μm.

overexpression of several functionally unrelated LD proteins (CG7900, dPlin1 or dPlin2) induces LDs suggests that LD accumulation is due to the shielding of LDs by LD proteins in photoreceptors. To test this hypothesis, we examined *Drosophila* expressing a fusion protein of GFP bound to the LD-binding domain (LD$^{BD}$::GFP) of the Nesprin family protein Klarsicht, that is required for the intracellular transport of LDs in *Drosophila* embryos [56,57]. Notably, photoreceptor-specific expression of LD$^{BD}$::GFP induced an accumulation of LDs (Figs 5E, 5F and S7), similar to that observed with *dPlin* or *CG7900* overexpression using an available *UAS-CG7900* transgenic line. Collectively, these results indicate that the shielding of LDs by LD proteins promotes LD accumulation in neurons, presumably by keeping away an active lipase or by stabilizing their structure.

We next examined the consequence of co-expression of *CG7900* or LD$^{BD}$::GFP and αSyn on LD accumulation. We found that αSyn expression enhanced synergistically the accumulation of LDs induced by *CG7900* or LD$^{BD}$::GFP (Fig 5E and 5F). These last results further support the hypothesis that distinct LD proteins by shielding the LDs from lipase, initiate LD accumulation, providing the appropriate surface for αSyn to bind and enhance LD accumulation.

To further characterize the effect of *CG7900* overexpression on LDs, we took advantage of a *UAS-αSyn$^{A53T}$* transgenic line, in which the insertion of αSyn$^{A53T}$ occurs in the promoter region of *CG7900* (S8A Fig, chromosomal position 3R-48; see Materials and Methods). As shown in S8B Fig, GAL4-mediated transcription of *UAS-αSyn$^{A53T}$* using the *GMR-GAL4* driver resulted in co-overexpression of *CG7900* and αSyn$^{A53T}$ (S8B Fig). This also resulted in the accumulation of LDs in photoreceptors, as visualized by BODIPY staining and transmission electron microscopy (S8C and S8D Fig). Knockdown of *CG7900* in the αSyn$^{A53T}$-*CG7900* transgenic line abolished the accumulation of LDs (S8E and S8F Fig), indicating that in these conditions as well, CG7900 is required for LD accumulation in photoreceptors. Interestingly, lipidomic analyses of retinas revealed that only TGs were enriched, but no other lipid and sterol content, in flies with pan-retinal expression of αSyn$^{A53T}$-*CG7900* compared to wild type αSyn (αSyn$^{WT}$) or control flies (S9 Fig). This suggests that LDs accumulating in photoreceptors contain TG and not sterol esters. Consistent with this, LD staining with BODIPY was abolished by overexpressing the TG lipase Bmm in αSyn$^{A53T}$- *CG7900* expressing flies (S10A and S10B Fig). These results demonstrate that TG is the major lipid in ectopic LD-induced by CG7900 and αSyn$^{A53T}$. Collectively these results support the fact that LD-binding proteins can act synergistically with αSyn to induce LD accumulation in photoreceptor neurons.

Finally, we considered the possibility that lipophagy could regulate LD levels in photoreceptors co-expressing αSyn and LD proteins. To evaluate lipophagy, we first looked for autophagic vacuoles containing LDs as previously reported [58]. For that, we examined if the autophagy mCherry::Atg8a reporter construct [59] co-localized with LDs in photoreceptors expressing αSyn$^{A53T}$-*CG7900*. We observed that BODIPY-labelled LDs, did not co-localize with the few mCherry::Atg8a punctae observed in photoreceptors (S11A Fig). Furthermore, there was no apparent difference in the accumulation of mCherry::Atg8a punctae in photoreceptor expressing αSyn$^{A53T}$-*CG7900* compared to wild type retina (S11B Fig). We also examined Ref(2)P protein accumulation, which is used as proxy for the autophagy flux [60]. We

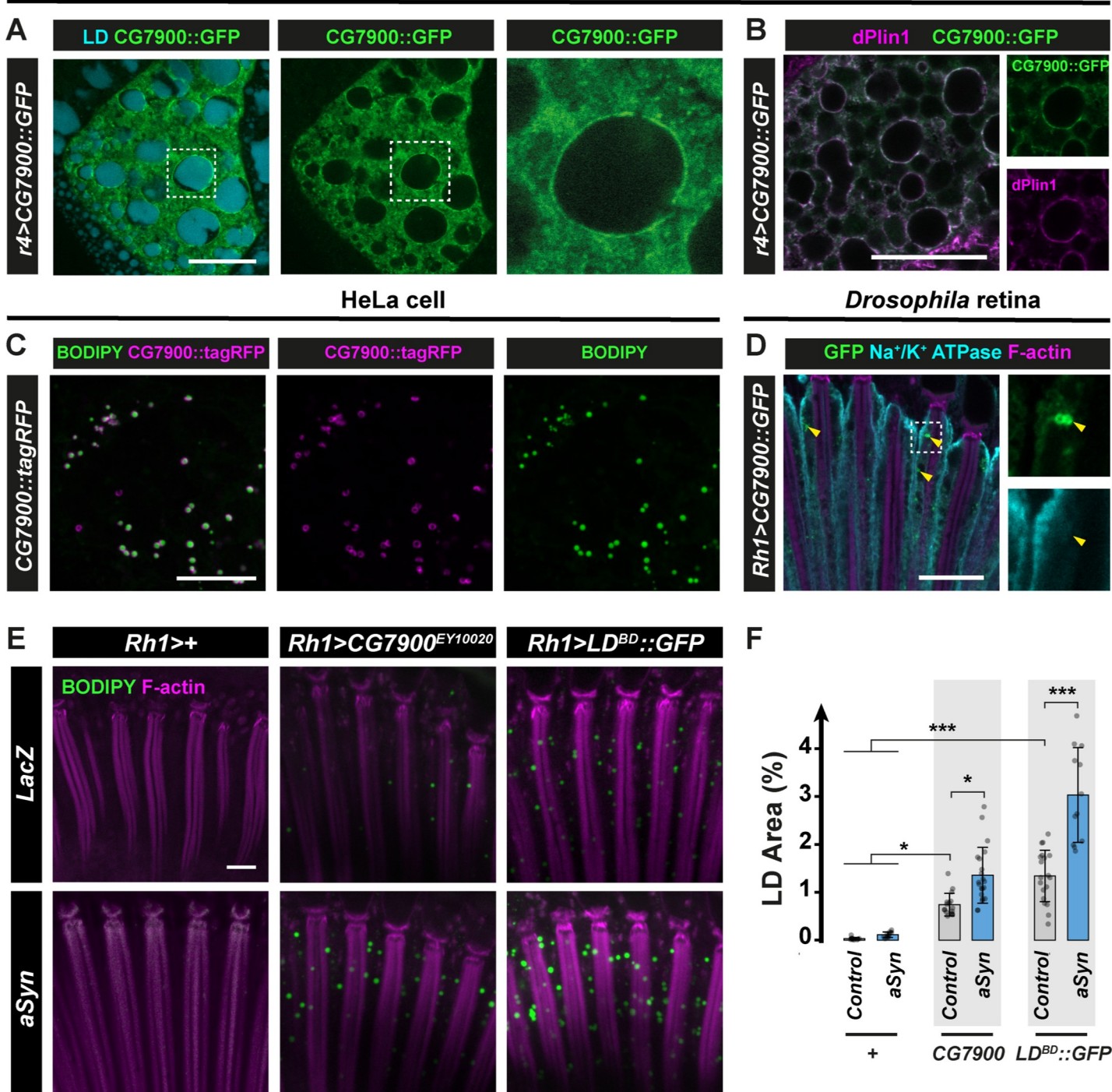

**Fig 5. CG7900 binds LDs in *Drosophila* tissues and promotes LD accumulation in photoreceptors. (A)** Immunostaining of whole mount *Drosophila* larval fat body expressing CG7900::GFP (*r4-GAL4*). CG7900::GFP is in green and LDs, stained with AUTOdot, are in cyan. Scale bar, 20μm. **(B)** Immunostaining of whole mount *Drosophila* larval fat body expressing CG7900::GFP (*r4-GAL4*). CG7900::GFP is in green and LDs are labeled with anti-dPlin1 (magenta). Scale bar, 20μm. **(C)** LD staining of HeLa cell transfected with *CG7900::tagRFP*. CG7900::tagRFP is in magenta and LDs are in green (BODIPY). Scale bar, 10 μm. **(D)** Immunostaining of whole-mount retinas from flies expressing CG7900::GFP in photoreceptor neurons (*Rh1-GAL4*). Photoreceptor plasma membranes are in cyan (anti-Na⁺/K⁺ ATPase) and rhabdomeres are in magenta (phalloidin-rhodamine). Arrowheads indicate ring shaped GFP positive staining in photoreceptors. Scale bar, 10 μm. **(E)** LD staining of whole-mount retina from flies with photoreceptor neuron-specific expression (*Rh1-GAL4*) of *LacZ* (control), *CG7900^{EY10020}* (EP-[UAS] insertion, EY10020), Klarsicht LD-binding domain (LD^{BD}::GFP), alone or in conjunction with αSyn. LDs are shown in green (BODIPY) and photoreceptor rhabdomeres in magenta (phalloidin-rhodamine). Scale bar, 10 μm. **(F)** Quantification of LD area from the images shown in (E). Mean ± SD. * $p<0.05$, *** $p<0.001$ by ANOVA with Tukey's HSD test.

found no difference in Ref(2)P in photoreceptor expressing $\alpha Syn^{A53T}$-CG7900 compared to wild type retina (S11C Fig). Collectively, these results suggest that autophagy, which is activated at basal levels in wild type retina [60,61], is not affected by $\alpha Syn^{A53T}$-CG7900 expression and that lipophagy is absent in these conditions. Next, we asked if the inhibition or the activation of autophagy could lead to a difference in the level of LDs in wild type or $\alpha Syn^{A53T}$-CG7900 expressing photoreceptors. For that we used *Atg8a* mutant flies and a UAS transgenic line to overexpress Atg1 [62]. We quantified LDs in $Atg8a^{KG07569}$ mutant photoreceptors (S11D and S11E Fig) and photoreceptors over-expressing *Atg1* (S11F and S11G Fig). We found no difference in the accumulation of LDs induced by the expression of $\alpha Syn^{A53T}$-CG7900 in those conditions compared to control. Collectively, our results indicate that lipophagy is not required for the regulation of LDs induced by the co-expression of $\alpha Syn^{A53T}$ and *CG7900*.

## LDs promote αSyn resistance to proteinase K

In human neurons, αSyn aggregation is a multi-step process involving accumulation of misfolded αSyn, a process that renders αSyn resistant to mild proteolysis using proteinase K [36,63]. Therefore, we investigated whether αSyn–LD interactions might influence the physical state/structure of αSyn in *Drosophila* photoreceptor neurons. αSyn resistance to proteinase K was first evaluated by western blot analysis of protein extracts from flies overexpressing $\alpha Syn^{WT}$ or *dPlin2* and $\alpha Syn^{WT}$. We observed that $\alpha Syn$ from 30-day-old transgenic flies was more resistant to proteinase K digestion with the concomitant expression of dPlin2 in which LDs accumulate (Fig 6A and 6B), correlating LD abundance and aberrant αSyn folding and/or aggregation. To confirm this link, we performed proteinase K-resistance assays on $\alpha Syn^{A53T}$-CG7900 from 30-day-old flies depleted of LDs by co-expression of Bmm lipase in photoreceptor neurons. Indeed, depletion of LDs significantly reduced $\alpha Syn^{A53T}$ resistance to proteinase K digestion (Fig 6C and 6D), This indicates that LD abundance had a crucial influence on αSyn proteinase K-resistance. Taken together, these results support our conclusion that LD accumulation enhances the resistance of αSyn to proteinase K.

## Discussion

In this study, we investigated the mechanisms that regulate LD homeostasis in neurons, the contribution of αSyn to LD homeostasis, and whether αSyn–LD binding influences the pathogenic potential of αSyn. We found that expression of the LD proteins, dPlin1 and dPlin2, CG7900 or of the LD-binding domain of Klarsicht increased LD accumulation in *Drosophila* photoreceptor neurons and that this phenotype was amplified by co-expressing the PD-associated protein αSyn. Transfected and endogenous αSyn co-localized with PLINs on the LD surface in human neuroblastoma cells, as demonstrated by confocal microscopy and PLA assays. Neuronal accumulation of LDs was not dependent on the canonical enzymes of TG synthesis (Mdy, dFatp), Bmm/dATGL-dependent lipolysis or lipophagy inhibition. One possible explanation for LD accumulation is that LD proteins inhibit an unknown lipase in *Drosophila* photoreceptor neurons. Finally, we observed that LD accumulation in photoreceptor neurons was associated with increased resistance of αSyn to proteinase K digestion, suggesting that LD accumulation might promote αSyn misfolding, an important step in the progression towards PD. Thus, we have uncovered a potential novel role for LDs in the pathogenicity of αSyn in PD.

Our understanding of the mechanisms of LD homeostasis in neurons under physiological or pathological conditions is far from complete. Neurons predominantly synthesize ATP through aerobic metabolism of glucose, rather than through FA β-oxidation, which likely

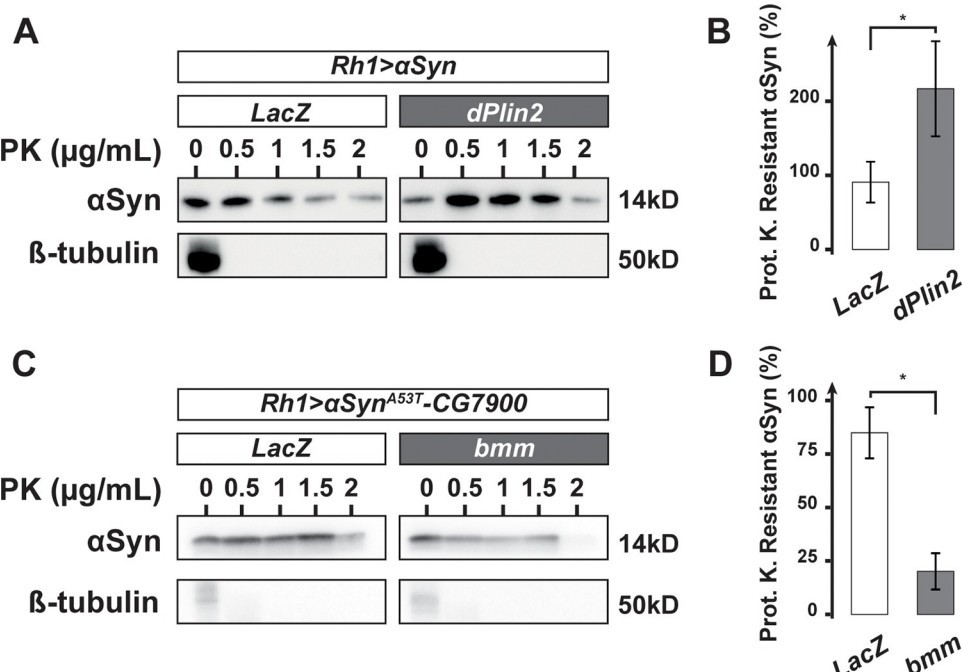

**Fig 6. LDs promote αSyn resistance to proteinase K in *Drosophila* photoreceptors.** **(A)** αSyn proteinase K-resistance assay. Lysates of the heads of 30-day-old flies with photoreceptor neuron-specific expression of *αSyn^WT* and either *LacZ* (control) or dPlin2::GFP were digested with the indicated concentrations of proteinase K and then immunoblotted for αSyn or β-tubulin (loading control). **(B)** Quantification of proteinase K-resistant αSyn, as analyzed in (A). Resistance is expressed as the ratio of αSyn detected after treatment with 1 μg/mL of proteinase K relative to the untreated sample. Mean ± SD. * p<0.05 by student test. **(C)** αSyn proteinase K-resistance assay. Lysates of the heads of 30-day-old flies with photoreceptor neuron-specific expression (*Rh1-GAL4*) of *αSyn^A53T*-CG7900* and either *LacZ* (control) or *bmm* lipase were digested with the indicated concentrations of proteinase K and then immunoblotted for αSyn or β-tubulin (loading control). **(D)** Quantification of proteinase K-resistant αSyn, as analyzed in (B). Resistance is expressed as the ratio of αSyn detected after treatment with 2 μg/mL of proteinase K relative to the untreated sample.

explains the relative scarcity of LDs in neurons compared with glial cells [64]. Here we used the *Drosophila* adult retina that is composed of photoreceptor neurons and glial cells to explore the mechanism regulating LD homeostasis in the nervous system. The canonical mechanisms regulating TG turnover and LD formation are dependent on evolutionary conserved regulators of lipogenesis and lipolysis in the fly adipose tissue, called fat body, or in other non-fat cells, such as glial cells [6,10]. Indeed, we and others showed that *de novo* TG-synthesis enzymes Dgat1/Mdy and dFatp, are required for LD biogenesis in the fat body and glial cells [7,8,10,25,26]. This is in contrast to dPlin-induced neuronal accumulation of LDs (this study), which occurs through a mechanism, independent of Mdy- and dFatp-mediated *de novo* TG synthesis. One possibility is that LD biogenesis depends on Dgat2 in neurons. However, the fact that there are three Dgat2 paralogs encoded by the fly genome [6] and that no triple mutant is available, precluded its functional analysis in the current study.

The evolutionarily conserved and canonical TG lipase Bmm, otholog of mammalian adipose triglyceride lipase (ATGL) regulates lipolysis in the fat body [14]. Here, we show that Bmm regulates LD abundance in glial cells but not in photoreceptor neurons. Interestingly, in both *bmm*-mutant *Drosophila* (this study) and ATGL-mutant mice [65], neurons do not accumulate LDs. This suggests the existence of an unknown and possibly cell type specific lipase regulating the degradation of LDs in neurons. This is supported by the fact that the

overexpression of dPlins proteins, which are known inhibitors of lipolysis, promotes LD accumulation in photoreceptor neurons. In further support of a neuron-specific TG lipase, the human hereditary spastic paraplegia gene DDHD2, a member of the iPLA1/PAPLA1 family, was proposed to be the main lipase regulating TG metabolism in the mammalian brain [66]. A recent study, showed that Bmm plays a role in the somatic cells of the gonad and in neurons to regulate systemic TG breakdown [67]. The authors also suggested that Bmm may play a role in regulating LD turnover in neurons, although this was not directly tested in this study. Our results using *bmm* knock-down and *bmm* mutants do not support a role of Bmm in the regulation of LD accumulation in photoreceptor neurons. However, we cannot exclude the possibility that Bmm would be required in a subpopulation of neurons to regulate LD content but this would require further analyses. Finally, we cannot exclude the possibility that the overexpression of LD proteins, such as dPlins but also CG7900 or the Klarsicht lipid-binding domain promotes LD accumulation by shielding and stabilizing LDs rather than limiting the access of lipases to LDs. Indeed, stabilization of LDs could well be an ancestral function of PLINs, as reported for yeast and *Drosophila* adipose tissue [7,40,68]. Thus, inhibiting lipolysis and/or stabilizing LDs, allows the formation of LDs, which would be otherwise actively degraded in photoreceptor neurons. This opens avenues to further study LD homeostasis but also their pathophysiological role in diseases of the nervous system.

Earlier studies have observed the accumulation of LDs in cellular models of PD. For example, LDs form in SH-SY5Y cells exposed to 1-methyl-4-phenyl-1,2,3,6-tetrahydropyridine (MPTP), a dopaminergic neurotoxin prodrug that causes PD-like symptoms in animal and cellular models [69]. In addition, studies in yeast, rat dopaminergic neurons, and human induced pluripotent stem cells have proposed that αSyn expression induces lipid dysregulation and LD accumulation, but the underlying mechanisms remained unclear [38,39,70]. Low levels of αSyn accumulation were hypothezised to perturb lipid homeostasis by enhancing unsaturated FA synthesis and the subsequent accumulation of DGs and TGs. In the present study, we showed that αSyn expression alone did not enhance the accumulation of LDs but instead required concomitant overexpression of a LD protein. Moreover, αSyn expression alone had no effect on DG, TG, or LD content in *Drosophila* photoreceptor neurons, which indicates that αSyn-induced LDs are not driven by increased TG biosynthesis in this cellular context. Instead, the fact that endogenous αSyn and PLIN3 proteins co-localized at the LD surface in human neuroblastoma cells, suggests that LD-associated αSyn have a direct physiological function in promoting neutral lipid accumulation by inhibiting lipolysis. This hypothesis is supported by experiments in HeLa cells transfected with αSyn, loaded with fatty acids, in which the overexpression of αSyn protects LDs from lipolysis [37].

Our results show that LDs contribute to αSyn conversion to proteinase K resistant forms, which indicates that LDs may be involved in the progression of PD pathology. This is an apparent discrepancy with the results in Fanning et al. (2019), in which LDs protect from lipotoxicity cells expressing αSyn [38]. In this study the authors used cellular models including yeast cells, and rat cortical neuron primary cultures exposed or not to oleic acid. In such cellular context, they propose that αSyn induces the accumulation of toxic diacylglycerol (DG), which is subsequently converted to TG and sequestered into LDs [38]. LDs are thus protective by allowing the sequestration of toxic lipids. In our fly retina study, αSyn expression did not induce TG accumulation. In the *Drosophila* nervous system, toxic DG may not reach sufficient level to promote photoreceptor toxicity. Interestingly, this difference allowed us to study the binding of αSyn to LD and examine their contribution to pathological conversion of αSyn, which was not explored in Fanning *et al.* [38]. Indeed, our results suggest an alternative but not mutually exclusive role for LDs in promoting αSyn misfolding and conversion to a proteinase K-resistant form. The increased LD surface could provide a physical platform for αSyn

deposition and conversion. In support of this hypothesis, it was previously proposed that αSyn aggregation is facilitated in the presence of synthetic phospholipid vesicles [71]. Thus, our results point to a direct role of LDs on αSyn resistance to proteinase K digestion.

We showed that the accumulation of LD proteins, such as dPlins, is a prerequisite for the increased LD accumulation induced by αSyn in neurons. This raises the possibility that some physiological or pathological conditions will favor the expression and/or accumulation of LD proteins, which triggers the neuronal accumulation of LDs. Interestingly, it was proposed that age-dependent accumulation of fat and dPlin2 is dependent on the histone deacetylase (HDAC6) in *Drosophila* [72]. Moreover, an accumulation of LD-containing cells (lipid-laden cells), associated with PLIN2 expression, was observed in meningeal, cortical and neurogenic brain regions of the aging mice [73]. Finally, a recent expression study on all human perlipin proteins (PLIN1-5), found that PLIN2 accumulates, particularly in neurons, in brains of old subjects and of patients with Alzheimer disease [74]. As an alternative putative mechanism regulating LD level, it was shown that targeted degradation of PLIN2 and PLIN3 occurs by chaperone-mediated autophagy (CMA) [75]. Thus, in aging tissue with decreased HDAC6 or reduced basal CMA, the accumulation of PLINs may initiate LD accumulation, hence favoring αSyn-induced LD production. In our hands, mutations in the central autophagy gene *Atg8* did not lead to LD accumulation in *Drosophila* retina. Thus a more systematic analysis will be required to identify the proteolytic mechanisms regulating dPlins degradation and LD accumulation in the aged *Drosophila* nervous system.

Based on a combination of our results and these observations, we propose a model of LD homeostasis in healthy and diseased neurons (Fig 7). In healthy neurons, relatively few LDs are detected due to a combination of low basal rate of TG synthesis, active lipolysis and limited LD shielding capacity. In pathological conditions such as PD, possibly in combination with an age-dependent ectopic fat accumulation and Plin proteins increased expression [72,76], αSyn and Plins could cooperate to limit lipolysis and promote the accumulation of LDs in neurons. This could set a vicious cycle in which αSyn enhances Plin-dependent LD stabilization, which, in turn, would increase αSyn conversion to a proteinase K-resistant form, culminating in αSyn aggregation and formation of cytoplasmic inclusion bodies. Collectively, our results raise the possibility that αSyn binding to LDs could be an important step in the pathogenesis of PD.

## Material and methods

### Fly stocks

All flies used for this study were raised on regular yeast medium at 25˚C on a 12h light/dark cycle. The fly stocks were obtained as follows. *UAS-αSyn* (third chromosome insertion BL8146, used throughout the paper unless it is specified), *UAS-αSyn$^{A53T}$-CG7900* (BL8148), *UAS-mdy-RNAi* (BL65963), *UAS-bmm-RNAi* (BL25926), *UAS-GFP* [77], *UAS-GFP-shRNA* (BL41555), *sGMR-GAL4, Rh1-GAL4* (BL8688) [46], *54C-Gal4* (BL27328) [10], *UAS-CG7900* (EY10020, BL17633), *UAS-LacZ* (BL1777), *r4-GAL4* (BL33832), *Atg8a$^{KG07569}$* (BL14639) and *UAS-Atg1* (BL51654) were from Bloomington *Drosophila* Stock Center. *UAS-dFatp-RNAi* (100124), *UAS-LacZ-RNAi* (51446) and *UAS-CG7900-RNAi* (101025) were from Vienna *Drosophila* Resource Center. *UAS-dPlin1::GFP* [7], *UAS-dPlin2::GFP* [20] (second and third chromosome insertions), *UAS-bmm* as well as *bmm$^1$* [14] and *mdy$^{QX25}$* [8] mutant flies were provided by R.P. Kühnlein (University of Graz); *UAS-LD$^{BD}$-GFP* was provided by M. Welte [57], *UAS-αSyn* (second chromosome insertion) was provided by M.B. Feany, *UAS-mCherry::Atg8a* (3$^{rd}$ chromosome insertion) was provided by I. Nezis [59].

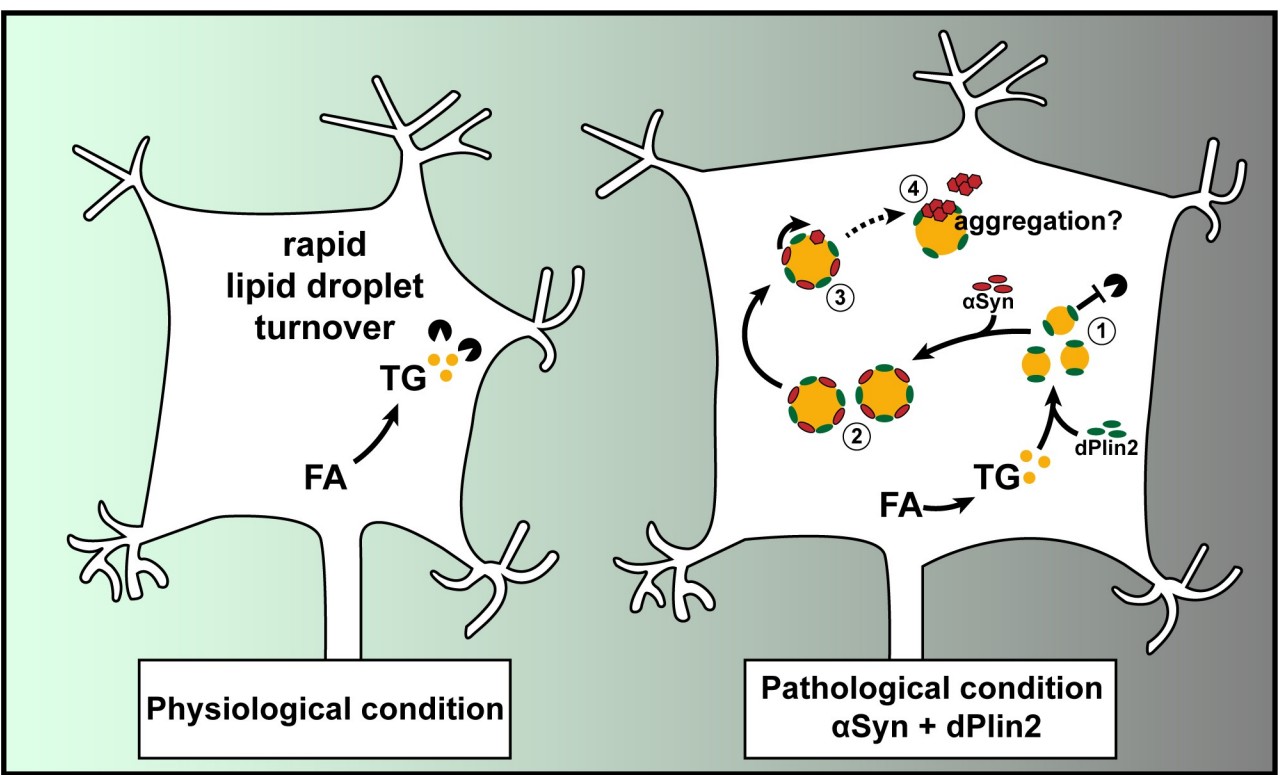

**Fig 7. Model of the reciprocal interactions between αSyn and LDs in PD.** Under normal physiological conditions, neurons contain relatively few LDs. We propose the following scenario under pathological conditions associated with elevated levels of αSyn and dPlin2: **(1)** increased dPlin2 limits the action of neuronal lipases and promotes LD accumulation; **(2)** αSyn binds to the expanding dPlin2-positive LDs, which further increases LD accumulation; **(3)** αSyn is converted to a proteinase K-resistant form(s) on the surface of LDs; and **(4)** the aberrant form of αSyn may aggregate at the surface of or in close proximity to LDs, leading to formation of cytoplasmic inclusion bodies [87].

FRT mediated d*Fatp* photoreceptor clones were generated as previously described [77] by crossing the following genotypes: *Rh1-GAL4*, *ey-FLP; FRT40A tdTomato::NinaC/ CyO;* and *FRT40A dFatp^{k10307}/CyO; UAS-dPlin1::GFP*.

### Generation of *UAS-CG7900::GFP* transgenic line and of *pCMV CG7900::TagRFP* vector

For the *UAS-CG7900::GFP* transgenic line, CG7900 cDNA was cloned (NotI/ BamHI) into a pJFRC2 vector in frame with the eGFP coding sequence [78]. Best Gene, Inc (CA, USA) generated transgenic lines using PhiC31 integrase mediated transgenesis. The vector DNA was injected in embryos carrying attP40 docking sites. For the pCMV CG7900::TagRFP vector, CG7900 cDNA was cloned (NheI/ XbaI) into a pTagRFP vector (Evrogen) upstream and in frame with the tagRFP coding sequence.

### Cell culture

The human neuroblastoma cell line SH-SY5Y was obtained from T. Baron (ANSES, Lyon, France) and transfected with 1 μg of a pcDNA3.1 vector containing human αSyn cDNA (provided by T. Baron, Anses, Lyon, France) using Effecten transfection reagent (Qiagen). Positive clones were selected using geneticin and cultured in Dulbecco's modified Eagle's medium (DMEM/F-12, Gibco) supplemented with 4.5 g/L D-glucose, 10% fetal bovine serum, 100 U/

mL penicillin, and 100 g/mL streptomycin at 37˚C. Cells were passaged when they reached 70–80% confluence. HeLa cells were obtained from ANIRA CelluloNet (SFR, Biosciences, Lyon, France). HeLa cells were transfected with 1 μg of a pTagRFP vector containing *CG7900* cDNA using the same transfection and selection protocoles as SH-SY5Y cells.

## LD staining

Unless otherwise stated, experiments were performed using 20-day-old female flies. Flies were sedated on ice, decapitated, and the retinas were dissected in a drop of HL3 medium [79]. Whole-mount retinas were fixed in 4% paraformaldehyde (PFA), briefly washed in PBS supplemented with 0.1% Triton X-100 (PBS-T), and incubated overnight at 4˚C in LD fluorescent dye BODIPY 493/503 (D3922, Molecular Probes) diluted in PBS-T supplemented with 1:400 phalloidin-rhodamine (R415, Molecular Probes) to label F-actin containing rhabdomeres. The retinas were rinsed once in PBS-T and then mounted on a bridged slide in Vectashield medium. Samples were examined on a Zeiss LSM800 at the LYMIC-PLATIM–Imaging and Microscopy Core Facility of SFR Biosciences (UMS3444), Lyon, France.

For larval fat body, wandering L3 larvae fat body were dissected, fixed in 4% PFA, briefly washed in PBS supplemented with 0.1% Triton X-100 (PBS-T), and incubated overnight at 4˚C in LD fluorescent dye AUTOdot 1:500 (Abgent, #SM1000b) [80].

## BODIPY quantification

Retina images were acquired on a Zeiss LSM800 confocal microscope as 16-bit stacks and processed for quantification using ImageJ software [81]. Images were first filtered for noise using Gaussian Blur 3D (σ = 1) and projected along the Z-axis. LDs were identified using the Otsu thresholding algorithm. The area of BODIPY staining was measured and divided by the total retinal area as previously described [10].

## Immunohistochemistry

Flies were sedated on ice, decapitated, and retinas were dissected in a drop of HL3 medium [79] supplemented with D-glucose (120 mM). Whole-mount retinas were fixed in 4% PFA and permeabilized in PBS supplemented with 0.5% Triton X-100 and 5 mg/mL BSA. Mouse anti-Na+/K+ ATPase α-subunit (1:400, a5, DSHB), rabbit anti-GFP (1:400, A6455, Invitrogen), mouse anti-αSyn (1:500, sc-12767, Santa Cruz Biotechnology), rabbit anti-Ref(2)P 1:1000 [82] or rabbit anti-dPlin2 rabbit 1:2000 [20], and rabbit anti-dPlin1 1:2000 [7] (a gift from R.P. Kühnlein) primary antibodies were diluted in blocking solution (PBS 1X, 0.5% Triton X-100, 5 mg/mL BSA and 4% Normal Goat Serum) and incubated with the retinas overnight at 4˚C. The samples were then washed and incubated overnight at 4˚C in blocking solution containing Alexa Fluor-conjugated anti-mouse Alexa488, anti-rabbit Alexa488, or anti-mouse Alexa647 secondary antibodies together with 1:400 phalloidin-rhodamine (R415, Molecular Probes) to label F-actin.

SH-SY5Y cells were fixed with 4% PFA for 15 min and permeabilized with PBS containing 5% BSA and 0.05% saponin for 15 min. Cells were then incubated with mouse anti-αSyn (1:2000, sc-12767, Santa Cruz Biotechnology) and rabbit anti-PLIN3 (1:500, NB110, Novus Biologicals) primary antibodies at room temperature for 1 h. The cells were then washed and incubated with BODIPY 493/503 (D3922, Molecular Probes) and Alexa Fluor-conjugated anti-rabbit Alexa488/Alexa546 or anti-mouse Alexa647 secondary antibodies. Nuclei were counterstained with 1 μg/mL DAPI. Slides were mounted in Mowiol 4–88 (Sigma-Aldrich) and imaged with a Zeiss LSM800 confocal microscope.

## Proximity ligation assay (PLA)

PLAs were performed using Duolink PLA kits (Sigma) according to the manufacturer's instructions. Briefly, cells were fixed in 4% PFA and incubated with mouse anti-αSyn (1:2000, sc-12767, Santa Cruz Biotechnology) and rabbit anti-PLIN3 (1:500, NB110, Novus Biologicals) antibodies diluted in PBS containing 5% BSA and 0.05% saponin. The cells were then incubated with Duolink probes (anti-rabbit plus, DIO88002 and anti-mouse minus DIO82004). The PLA signal was revealed using the red Duolink In Situ Detection Reagent (DUO92008) and the cells were stained with BODIPY 493/503 (D3922, Molecular Probes). Nuclei were counterstained with DAPI in Mowiol mounting medium.

## Mapping of P{UAS-αSyn$^{A53T}$} insertion site

P{UAS-αSyn$^{A53T}$} genomic localization was mapped using the Splinkerette protocol for mapping of transposable elements in *Drosophila* [83]. Briefly, genomic DNA was isolated from one fly (stock BL8148) and digested using BstYI. DNA fragments containing the P-element flanking regions were then amplified using primers specific for pCaSpeR based P-element. The resulting DNA fragments were sequenced and mapped to the *Drosophila* genome using the BLAST platform (3R: 8,090,590..8,094,096).

## Transmission electron microscopy (TEM) of *Drosophila* eyes

TEM sample preparation was performed as previously described [10]. Briefly, *Drosophila* eyes were fixed in 0.1 M cacodylate buffer supplemented with 2.5% glutaraldehyde and 2 mM $CaCl_2$ for 16 h at 4°C. After rinsing with 0.1 M cacodylate, the tissues were contrasted by incubation in 1% $OsO_4$ in 0.1 M cacodylate buffer for 2 h at room temperature. Tissues were then dehydrated in acetone and mounted in 100% epoxy resin (Epon 812). After resin polymerization, samples were sliced into 60 nm sections, stained with lead citrate, and examined with a Philips CM120 TEM operating at 80 kV.

## Lipid extraction and quantification by shotgun mass spectrometry

Ten retinas per biological sample were homogenized twice for 5 min each with 1 mm zirconia beads in 300 μl isopropanol using a cooled Tissuelyzer II at 30 Hz. The homogenate was evaporated in a vacuum desiccator to complete dryness, and lipids were extracted as described [84,85]. After evaporation, the samples were reconstituted in 300 μL 1:2 $CHCl_3$:MeOH. To quantify sterols, 200 μL aliquots of lipid extracts were evaporated and acetylated with 300 μL 2:1 $CHCl_3$:acetyl chloride for 1 h at room temperature (method modified from [86]). After evaporation, sterol samples were reconstituted in 200 μL 4:2:1 isopropanol:MeOH:$CHCl_3$ with 7.5 mM ammonium formate (spray solution). For sterol and lipidome measurements, samples were diluted 1:1 with spray solution. Mass spectrometric analysis was performed as described [84].

## RNA extraction and qRT-PCR

Total RNA was extracted from three sets of 10 *Drosophila* heads using TRI-Reagent (T9424, Sigma) and RNA was reverse transcribed using an iScript cDNA Synthesis Kit (Bio-Rad) according to the manufacturers' instructions. Quantitative PCR reactions were carried out using FastStart Universal SYBER Green Master mix (Roche Applied Science) on a StepOne-Plus system (Applied Biosystems). Primer efficiency (E) was assessed using serial dilutions of cDNA preparations. Standard curves were used to determine the relationship between PCR cycle number (Ct) and mRNA abundance. Relative mRNA quantity (Qr) was calculated as:

$Qr = E^\wedge Ct_{Rp49}\text{-}Ct_{target}$. Qr values were then normalized to control genotype. Experiments were performed using the following primers: *CG7900*: 5′-CTGCTCACTCTCAGCGTTCAG-3′ and 5′-ATATGTGCGAACCAACTCCAC-3′; *Rp49*: 5′-ATCGTGAAGAAGCGCACCAAG-3′ and 5′-ACCAGGAACTTCTTGAATCCG-3′.

## Proteinase K-resistance assay

Fly heads were homogenized in lysis buffer (50 mM Tris-HCl pH 7.5, 5 mM EDTA, 0.1% NP40, 1 mM DTT, and 1% Protease Inhibitor Cocktail), incubated for 1 h at 25˚C, and centrifuged at 13,000 rpm for 1 min. Supernatants were collected and incubated for 30 min at 25˚C with proteinase K (0, 0.5, 1, 1.5, or 2 μg/mL). Denaturing buffer TD4215 4X was added to each sample, and proteins were separated in 4%–15% gradient acrylamide gels (Bio-Rad) and transferred to PVDF membranes (Millipore). PVDF membranes were fixed in 4% PFA and 0.01% glutaraldehyde in PBS for 30 min and then blocked in 3% BSA/0.1% Tween/PBS for 1h. Membranes were incubated with rabbit anti-α-Syn (MJFR1, ab138501, Abcam; 1:1000) or mouse anti-β-tubulin (T 6199, Sigma, 1:1000) primary antibodies overnight at 4˚C, washed, and incubated with horseradish peroxidase-conjugated anti-mouse IgG or anti-rabbit IgG (both from Pierce, 1:1000). After washing, membranes were incubated with SuperSignal West Dura Chemiluminescence Substrate (Thermo Scientific), and images were acquired using a Chemi-Doc MP system (Bio-Rad).

## Statistical analysis

Data are presented as the means ± standard deviation (SD) of three experiments unless noted. Statistical analyses were performed using R software. Differences between groups were analyzed by t-test or ANOVA and Tukey's HSD paired sample comparison test depending on the number of groups, as specified in the figure legends.

## Supporting information

**S1 Fig. The *Drosophila* retinal structure. (A)** Diagram of a *Drosophila* compound eye. The eye is composed of about 800 repeated units called ommatidia (hexagonal shapes). Longitudinal sections (shown in white) of the ommatidia allow visualization of the retinal cells that span the entire width of the retina. **(B)** Diagram of a longitudinal section of one ommatidium. Each ommatidium is composed of 8 photoreceptor neurons (light blue), each containing one rhabdomere (dark gray), and glial cells (also known as primary, secondary and tertiary retinal pigment cells; maroon, medium red and, light red respectively) that are juxtaposed to photoreceptor neurons from the apical to the basal retina. **(C)** Diagram of a cross-section of one ommatidium. In addition to the glial cells (2 primary, 6 secondary [medium red] and 3 tertiary [light red] pigment cells), each ommatidium contains 3 bristle cells (yellow) originating from the neuronal lineage. **(D)** Immunostaining of whole-mount retinas from flies expressing GFP in photoreceptor (*Rh1-GAL4*). Photoreceptor plasma membranes are in cyan (anti-Na$^+$/K$^+$ ATPase) and rhabdomeres are in magenta (phalloidin-rhodamine). GFP is visible only in the photoreceptor cytoplasm. Scale bar, 10μm.
(TIF)

**S2 Fig. dPlin2-induced LDs do not require canonical enzymes involved in TG synthesis (dFatp, Mdy) in photoreceptor neurons. (A)** LD staining of whole-mount retinas from flipase mediated *FRT40A dFatp^{k10307}* mutant clone in conjunction with expression of *dPlin2::GFP* in photoreceptors (*Rh1-GAL4*). LDs are in green (BODIPY), wild type photoreceptors are in magenta (*FRT40A tdTomato::NinaC*) and rhabdomeres are in grey (phalloidin-rhodamine

labeling of F-actin). LDs are visible in photoreceptors lacking *dFatp* (mutant clone surrounded by dashed line). Scale bar, 10 μm. **(B, C)** LD staining of whole-mount retinas from *mdy*<sup>QX25</sup> heterozygous and homozygous flies expressing *dPlin2*::*GFP* in photoreceptors (*Rh1-GAL4*). Scale bar, 10 μm.**(B)** LDs are in green (BODIPY), and rhabdomeres are in magenta (phalloidin-rhodamine). **(C)** Photoreceptor plasma membranes are in cyan (anti-Na$^+$/K$^+$ ATPase) and rhabdomeres are in magenta (phalloidin-rhodamine) dPlin2::GFP stains LDs as a ring shape in the photoreceptor cytoplasm (yellow arrowheads).
(TIF)

**S3 Fig. Loss of *bmm* function promotes LDs in glial cells but not in photoreceptor neurons. (A)** Immunostaining of whole-mount retinas from control (*w*$^{1118}$) or homozygous *bmm*$^1$ mutant flies. LDs are in green (anti-dPlin2), photoreceptor plasma membranes are in cyan (anti-Na$^+$/K$^+$ ATPase) and rhabdomeres are in magenta (phalloidin-rhodamine). Scale bar, 10 μm. **(B)** Immunostaining of whole-mount retinas from flies expressing RNAi targeting *bmm* lipase under the control of the pan-retinal driver *GMR-GAL4*. LDs are visible in green (anti-dPlin2), photoreceptor plasma membranes are in cyan (anti-Na$^+$/K$^+$ ATPase) and rhabdomeres are in magenta (phalloidin-rhodamine). Scale bar, 10 μm.
(TIF)

**S4 Fig. αSyn and dPlin2 synergize to induce LD accumulation in photoreceptor neurons. (A)** LD staining of whole-mount retinas from flies with photoreceptor neuron-specific expression of *LacZ* (control) or *αSyn*$^{WT}$ (two independent lines located on second [II] and third chromosome [III]; the line αSyn III is used elsewhere in the article) alone or in conjunction with *dPlin2*::*GFP*. LDs are shown in green (BODIPY) and photoreceptor rhabdomeres are shown in magenta (phalloidin-rhodamine). Scale bar, 10 μm. **(B)** Quantification of LD area from the images shown in (A). Mean ± SD. $^{***}$p<0.001, $^{**}$p<0.01 by ANOVA with Tukey's HSD test.
(TIF)

**S5 Fig. αSyn enhances dPlin1-induced LD accumulation in *Drosophila* photoreceptor neurons. (A)** LD staining of whole-mount retinas from flies expressing LacZ (control) or human αSyn alone or in conjunction with *dPlin1*::*GFP* in photoreceptor neurons (*Rh1-GAL4*). LDs are in green (BODIPY) and photoreceptor rhabdomeres are in magenta (phalloidin-rhodamine). Scale bar, 10 μm. **(B)** Quantification of LD area from images shown in (A). Mean ± SD. $^{***}$p<0.001 by ANOVA with Tukey's HSD test.
(TIF)

**S6 Fig. αSyn co-localizes with PLIN3 in human neuroblastoma cells. (A).** Proximity ligation assay between αSyn and PLIN3 in non-transfected SH-SY5Y cells. The PLA signal generated by close proximity of the two protein-bound primary antibodies shown in magenta is not visible when antibody against PLIN3 or αSyn are used separately. LDs are in green (BODIPY), and nuclei are counterstained with DAPI (cyan). Scale bars, 15 μm. **(B).** Quantification of the number of PLA foci per cell seen in (A). **(C).** Proximity ligation assay between αSyn and PLIN3 in transfected SH-SY5Y cells with *αSyn*$^{WT}$. The PLA signal generated by close proximity of the two protein-bound primary antibodies shown in magenta is not visible when antibody against PLIN3 or αSyn are used separately. LDs are in green (BODIPY), and nuclei are counterstained with DAPI (cyan). Scale bars, 15 μm. **(D).** Quantification of the number of PLA foci per cell seen in (C).
(TIF)

**S7 Fig. The expression of GFP-tagged LD-binding domain of Klarsicht induces LD accumulation in photoreceptor neurons.** Immunostaining of whole-mount retinas from flies with photoreceptor neuron-specific expression of the minimal LD-binding domain of Klarsicht fused to GFP (*UAS-LD$^{BD}$::GFP*) [57]. GFP localizes to the ring-shaped structures (yellow arrowheads) located between photoreceptor plasma membranes in cyan (anti-Na$^+$/K$^+$ ATPase) and rhabdomere in magenta (phalloidin-rhodamine). Scale bar, 10 μm. (TIF)

**S8 Fig. Characterization of the *P{UAS-αSyn$^{A53T}$}CG7900* transgenic line promoting LD accumulation in photoreceptors. (A)** Diagram of the genomic localization of the *P{UAS-αSyn$^{A53T}$}* transgene mapped using the Splinkerette protocol. The P-element carrying the upstream activating sequence (UAS) upstream of the coding sequence of human *αSyn$^{A53T}$* is inserted in the promoter region of *CG7900*. **(B)** RT-qPCR analysis of *CG7900* mRNA in heads of flies expressing *LacZ*, *αSyn$^{A53T}$-CG7900*, or *CG7900* (EP-[UAS] insertion, EY10020) under the control of the pan-retinal driver *GMR-GAL4*. mRNA levels are expressed as the mean ± SD of triplicates relative to the level in control (*GMR>LacZ*) flies. **(C)** LD staining of whole-mount retinas from flies with pan-retinal expression of *GFP* or *αSyn$^{A53T}$-CG7900*. LDs are shown in green (BODIPY) and photoreceptor rhabdomeres are in magenta (phalloidin-rhodamine). Scale bar, 20 μm. **(D)** TEM images of ommatidia cross-sections from 60-day-old flies with pan-retinal expression of *GFP* (top panel) or *αSyn$^{A53T}$-CG7900* (bottom panel). Each panel shows a representative cross-section of one ommatidium containing seven photoreceptors (false-colored blue) with central rhabdomeres (R) surrounded by retinal glial cells (false-colored orange). Yellow asterisks indicate LDs accumulating in the photoreceptor cytoplasm of flies expressing *αSyn$^{A53T}$-CG7900*. Scale bar, 2 μm. **(E)** LD staining of whole-mount retinas from flies expressing *αSyn$^{A53T}$-CG7900* in conjunction with *GFP-RNAi* or *CG7900-RNAi* in photoreceptors (*GMR-GAL4*). LDs are shown in green (BODIPY) and photoreceptor rhabdomeres are in magenta (phalloidin-rhodamine). Scale bar, 20 μm. **(F)** Quantification of LD area from the images shown in (E). Mean ± SD. \*\*\*p<0.001 by t-test. (TIF)

**S9 Fig. Expression of αSyn$^{A53T}$-CG7900 induces TG accumulation in *Drosophila* retina. (A)** Main lipid classes detected by shotgun mass spectrometric analysis of retinas from 20-day-old flies with pan-retinal expression of *LacZ* (control, gray bars), *αSyn$^{WT}$* (blue bars), or *αSyn$^{A53T}$-CG7900* (red bars). Lipid quantities are expressed as mole % within each species. Data show the mean ± SD of five biological replicates. Expression of *αSyn$^{A53T}$-CG7900* induces a significant accumulation of triacylglycerols (TG) in *Drosophila* retina. PC, phosphatidylcholine; PE, phosphatidylethanolamine; PA, phosphatidic acid; PI, phosphatidylinositol; PS, phosphatidyl serine; DG, diacylglycerol. **(B)** Sterol composition of retinas from 20-day-old flies with pan-retinal expression of LacZ (control, gray bars), *αSyn$^{WT}$* (blue bars), or *αSyn$^{A53T}$-CG7900* (red bars). Lipid quantities are expressed as mole % within each species. Data show the mean ± SD of 5 biological replicates. Erg, ergosterol; Bra, brassicasterol; Sit, sitosterol; Cam, campesterol; Cho, cholesterol; Zym, zymosterol; Lan, lanosterol and Sti, stigmasterol (TIF)

**S10 Fig.** Bmm promotes the degradation of αSyn$^{A53T}$-CG7900 induced-LDs in *Drosophila* photoreceptors (A) LD staining of whole-mount retinas from flies expressing αSynA53T-CG7900 alone or in conjunction with bmm lipase in photoreceptors (*Rh1-GAL4*). LDs are shown in green (BODIPY) and photoreceptor rhabdomeres are in magenta (phalloidin-rhodamine). Scale bar, 20 μm. (B) Quantification of LD area from the images shown in

(E). Mean ± SD. ***p<0.001 by t-test.
(TIF)

**S11 Fig. Lipophagy does not modulate αSyn^A53T^-CG7900 induced-LDs in *Drosophila* photoreceptors. (A)** LD staining of whole-mount retinas from flies expressing *mCherry::Atg8a* in conjunction with *LacZ* (Control) or *αSyn^A53T^-CG7900* in photoreceptors (*Rh1-GAL4*). LDs are shown in green (BODIPY), autophagosomes are in magenta (mCherry::Atg8a) and photoreceptor rhabdomeres are in cyan (phalloidin-rhodamine). Scale bar, 10 μm. **(B)** Quantification of mCherry::Atg8a area expressed in fold change compared to control from the images shown in (A). Mean ± SD. Ns, Non-significant by t-test. **(C)** Quantification of Ref(2)P punctae (anti-Ref(2)P labelling) of flies expressing *LacZ* (Control) or *αSyn^A53T^-CG7900* in photoreceptors (*Rh1-GAL4*). Expressed as fold change compared to LacZ controls. Mean ± SD. Non-significant by t-test. **(D)** LD staining of whole-mount retinas from *Atg8a^KG07569^* flies expressing *αSyn^A53T^-CG7900* under the control of pan-retinal driver (*GMR-GAL4*). LDs are shown in green (BODIPY), and photoreceptor rhabdomeres are in cyan (phalloidin-rhodamine). Scale bar, 10 μm. **(E)** Quantification of LD area from the images shown in (D). Mean ± SD. ns, not significant by ANOVA. **(F)** LD staining of whole-mount retinas from flies expressing *αSyn^A53T^-CG7900* alone or in conjunction with *Atg1* in photoreceptors (*Rh1-GAL4*). LDs are shown in green (BODIPY), and photoreceptor rhabdomeres are in cyan (phalloidin-rhodamine). Scale bar, 20 μm. **(G)** Quantification of LD area from the images shown in (F). Mean ± SD. ns, not significant by ANOVA.
(TIF)

## Acknowledgments

We thank the ARTHRO-TOOLS and LYMIC-PLATIM microscopy platforms of SFR Biosciences (UMS3444/CNRS, US8/INSERM, ENS de Lyon, UCBL) and the Centre Technologique des Microstructures CTμ at Lyon 1 University. We also thank Lisa Pichler in Graz for excellent technical assistance. We are gratefull to our colleagues and the stock fly centers for kindly sending fly stocks and antibodies.

## Author Contributions

**Conceptualization:** Victor Girard, Oskar Knittelfelder, Daan M. Van den Brink, Ronald P. Kühnlein, Nathalie Davoust, Bertrand Mollereau.

**Data curation:** Victor Girard, Florence Jollivet.

**Formal analysis:** Victor Girard, Florence Jollivet, Oskar Knittelfelder, Nathalie Davoust, Bertrand Mollereau.

**Funding acquisition:** Andrej Shevchenko, Ronald P. Kühnlein, Nathalie Davoust, Bertrand Mollereau.

**Investigation:** Victor Girard, Florence Jollivet, Marion Celle, Gilles Chatelain.

**Methodology:** Victor Girard, Florence Jollivet, Oskar Knittelfelder, Jean-Noel Arsac, Thierry Baron, Andrej Shevchenko.

**Project administration:** Nathalie Davoust, Bertrand Mollereau.

**Supervision:** Thierry Baron, Andrej Shevchenko, Ronald P. Kühnlein, Nathalie Davoust, Bertrand Mollereau.

**Visualization:** Victor Girard.

**Writing – original draft:** Victor Girard, Nathalie Davoust, Bertrand Mollereau.

**Writing – review & editing:** Victor Girard, Ronald P. Kühnlein, Nathalie Davoust.

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
