## [Decision Letter · Decision Letter 0]

8 Jul 2021

Dear Dr Mollereau,

Thank you very much for submitting your Research Article entitled 'A non-canonical lipid droplet metabolism regulates the conversion of alpha-Synuclein to proteolytic resistant forms in neurons of a Drosophila model of Parkinson disease' to PLOS Genetics.

The manuscript was seen by 2 reviewers who also had evaluated your earlier manuscript D-20-01524. Reviewers #1 and #2 of the current submission correspond to reviewers #1 and #3, respectively, for the earlier submission. As you will see, they are split in their enthusiasm for the work, with reviewer #2 raising a number of important concerns regarding interpretation and impact.

The manuscript and the reviews have now been discussed among members of the editorial board. Overall, we are interested to evaluate what we hope will be a final major revision to address key concerns raised by reviewer #2. Many of the points can be addressed by textual revisions but we caution that potential success of a revision will require additional experimentation.

If you decide to revise the manuscript for further consideration at PLOS Genetics, please aim to resubmit within the next 60 days, unless it will take extra time to address the concerns of the reviewers, in which case we would appreciate an expected resubmission date by email to plosgenetics@plos.org.

[LINK]

We are sorry that we cannot be more positive about your manuscript at this stage. Please do not hesitate to contact us if you have any concerns or questions.

Yours sincerely,

Gregory P. Copenhaver, Ph.D.

Editor-in-Chief

PLOS Genetics

Gregory Barsh

Editor-in-Chief

PLOS Genetics

Reviewer's Responses to Questions

**Comments to the Authors:**

Reviewer #1: This is a revised manuscript reporting that Lipid Droplet (LD) coating proteins synergize with alpha synucleins to promote LD accumulation in neurons, and that LDs protect alpha synucleins from proteolytic digestion, a property associated with alpha synuclein aggregation in human cells. Through the revision process, the authors have satisfactorily addressed points raised during the initial review cycle. Among others, they added new data showing that different sets of LD binding proteins have similar properties in synergizing with alpha synucleins. With the newly added data, the organization of the manuscript has significantly improved. Although the authors did not experimentally address the effect of LDs on alpha synuclein-induced neurodegeneration, they now highlight LD's effect on the pathological conversion of alpha synuclein to a proteinase K resistant form. Accordingly, they changed the title reflecting this focus.

This is a fine study and I support the publication of this manuscript.

I noticed a few grammar mistakes in the revised manuscript text, and I recommend fixing those errors prior to publication.

Reviewer #2: In their resubmitted manuscript, Girard et al. targeted most, yet not all of my comments and issues raised in the initial submission. Unfortunately, I find not all of the arguments or newly presented data fully convincing. While I appreciate the efforts made, I continued to ask myself whether the authors could make the claim presented in the title of the manuscript (the title changed now to: “A non-canonical lipid droplet metabolism regulates the conversion of alpha-Synuclein to proteolytic resistant forms in neurons of a Drosophila model of Parkinson disease”). The authors actually leave it in my opinion open what the “non-canonical lipid droplet metabolism” really means. They tested a (the?) very prominent lipase and two enzymes important for neutral lipid deposition. Yet, other processes, such as autophagy and more specifically lipophagy remain untouched (there is one remark that Atg8 perturbation did not affect the phenotype). Thus, a key aspect of the study – how the LDs form and potentially affect the alpha-synuclein function – remains unanswered. As this is a central point of the paper, this needs to be addressed before publishing the article can be considered at this level.

As outlined above, a key point is the hypothesis that an unknown lipase degrades LDs in neurons in a futile cycle and that this lipase is different from Brummer. The authors base this statement on the data shown in Figures 2 and S3. In order to test whether lipogenesis is required for the neuronal LD deposition, they co-express the LD-associated protein(s) and target / use mutants of the anabolic enzymes midway and FATP2, respectively. In the corresponding lipase loss-of-function experiments, however, they omit the LD protein co-expression. What is the rationale behind this? In neurons, there might be just not enough coating protein(s) available to stabilize the lipids in droplets without the presence of additional protein (which leaves the situation somewhat artificial; the supplemental data uses antibody staining to detect endogenous Lsd-2 which is indeed lacking in the neurons). Thus, the statement that a different lipase should be required for the build-up of droplets is not clear to me and only one possible explanation. I understand that the overexpression of the LD proteins alone results in the appearance of LDs; yet in case a lipase (brummer?) is involved, its knock-down should result in even larger LDs. The authors thus generate hypotheses (including the ill-defined “non-canonical lipid droplet metabolism”), which are not really fully supported by their data.

The authors mention that they tested a collection of lipogenic gene functions for their role in the LD deposition (information from the rebuttal letter). They should add this data to the manuscript given the appropriate controls (e.g. knock-down efficacy) are present to further strengthen their point. Also, the authors unfortunately decided against following my suggestion to provide fluorescent fatty acids to test for an importance of exogenous lipid supply for the build-up of the LDs, which I still think could have helped to better test the mechanism of the LD origin. Further, they include in their model exogenous fatty acids as source for the LD deposition but not really test this. As an alternative to my initially suggested experiment, the authors could also test what happens if the amount of circulating free fatty acids is increased e.g. by an overexpression of brummer in the fat body. Given that they already use brummer overexpression (Fig. S10), this should be a feasible experiment.

Concerning the “substantial additional characterization of CG7900 which demonstrates that it is a bonafide LD protein (Figure 5A and 5B)” (quoted from the rebuttal letter) I am not that convinced concerning the substantiality of characterization. The fat body is densely packed with LDs and accordingly, the admittingly circular signal might indicate LD localization. Yet, it could also be close apposition of e.g. the ER and to really show the LD localization I would expect e.g. a biochemical LD purification and western blotting to demonstrate the co-purification of the protein.

Taken together, I still think the study is lacking significant experiments to support the presented hypotheses and conclusions and thus I do not recommend its publication in PLoS Genetics at this point of time.

**Have all data underlying the figures and results presented in the manuscript been provided?**

Reviewer #1: Yes

Reviewer #2: Yes

PLOS authors have the option to publish the peer review history of their article (what does this mean?). If published, this will include your full peer review and any attached files.

Reviewer #1: **Yes: **Hyung Don Ryoo

Reviewer #2: No

---

## [Decision Letter · Decision Letter 1]

2 Nov 2021

Dear Dr Mollereau,

We are pleased to inform you that your manuscript entitled "Abnormal accumulation of lipid droplets in neurons induces the conversion of alpha-Synuclein to proteolytic resistant forms in a Drosophila model of Parkinson disease" has been editorially accepted for publication in PLOS Genetics. Congratulations!

Yours sincerely,

Gregory P. Copenhaver

Editor-in-Chief

PLOS Genetics

Gregory Barsh

Editor-in-Chief

PLOS Genetics

Comments from the reviewers (if applicable):

Reviewer's Responses to Questions

**Comments to the Authors:**

Reviewer #2: In their revised version of the manuscript, as well as the rebuttal letter, the authors fully addressed all my previous comments. I very much appreciate the addition of the new and supporting data as well as the edits in the text and figures. Thus, I support the publication of this manuscript.

**Have all data underlying the figures and results presented in the manuscript been provided?**

Reviewer #2: Yes

PLOS authors have the option to publish the peer review history of their article (what does this mean?). If published, this will include your full peer review and any attached files.

Reviewer #2: No

**Data Deposition**

http://datadryad.org/submit?journalID=pgenetics&manu=PGENETICS-D-21-00810R1

**Press Queries**

---

## [Editor Report · Acceptance letter]

12 Nov 2021

PGENETICS-D-21-00810R1 

Abnormal accumulation of lipid droplets in neurons induces the conversion of alpha-Synuclein to proteolytic resistant forms in a Drosophila model of Parkinson disease 

Dear Dr Mollereau, 

We are pleased to inform you that your manuscript entitled "Abnormal accumulation of lipid droplets in neurons induces the conversion of alpha-Synuclein to proteolytic resistant forms in a Drosophila model of Parkinson disease" has been formally accepted for publication in PLOS Genetics! Your manuscript is now with our production department and you will be notified of the publication date in due course.

With kind regards,

Katalin Szabo

PLOS Genetics

On behalf of:
